# A Novel Temperature Drift Compensation Algorithm for Liquid-Level Measurement Systems

**DOI:** 10.3390/mi16010024

**Published:** 2024-12-27

**Authors:** Shanglong Li, Wanjia Gao, Wenyi Liu

**Affiliations:** 1Key Laboratory of Micro/Nano Devices and Systems, Ministry of Education, North University of China, Taiyuan 030051, China; wanglipei2021@163.com (S.L.); 18810577682@163.com (W.G.); 2State Key Laboratory of Dynamic, Measurement Technology, North University of China, Taiyuan 030051, China

**Keywords:** ultrasonic impedance, ultrasonic sensors, temperature drift, temperature compensation

## Abstract

Aiming at the problem that ultrasonic detection is greatly affected by temperature drift, this paper investigates a novel temperature compensation algorithm. Ultrasonic impedance-based liquid-level measurement is a crucial non-contact, non-destructive technique. However, temperature drift can severely affect the accuracy of experimental measurements based on this technology. Theoretical analysis and experimental research on temperature drift phenomena are conducted in this study, accompanied by the proposal of a new compensation algorithm. Leveraging an external fixed-point liquid-level detection system experimental platform, the impact of temperature drift on ultrasonic echo energy and actual liquid-level height is examined. Experimental results demonstrate that temperature drift affects the speed and attenuation of ultrasonic waves, leading to decreased accuracy in measuring liquid levels. The proposed temperature compensation method yields an average relative error of 3.427%. The error range spans from 0.03 cm to 0.336 cm. The average relative error reduces by 21.535% compared with before compensation, showcasing its applicability across multiple temperature conditions and its significance in enhancing the accuracy of ultrasonic-based measurements.

## 1. Introduction

In traditional fields, real-time liquid-level detection and alarm in sealed or pressure vessels under harsh conditions are imperative [1]. Therefore, research on the accuracy of liquid-level sensors is crucial. There are primarily two types of liquid-level measurement techniques: invasive and non-invasive [2]. Invasive liquid-level measurement techniques require direct contact with the liquid for measurement, thus compromising the integrity of the measured container and making non-destructive testing impossible for large sealed containers in harsh environments [3]. Non-invasive liquid-level measurement techniques, on the other hand, do not directly involve contact with the container for liquid-level measurement, making them suitable for measuring liquid levels in large sealed containers. Ultrasonic detection technology represents a non-invasive liquid-level measurement technique capable of non-destructive testing, and it has gradually become the mainstream technology for liquid-level detection [4]. Ultrasonic sensors are primarily classified into three types: The first type involves transducers that both emit and receive signals [5]. The phase, frequency, and amplitude of the ultrasound change when interacting with reflective particles or bubbles in liquids or gases. The second type uses two separate transducers, one for transmission and one for reception [6]. The phase and amplitude of the ultrasound are altered based on the characteristics of the transmission medium, carrying information about it between the emitter and receiver. The third type employs a transducer solely as a receiver [7], passively collecting acoustic energy generated by the object under measurement and providing general information about the object. The detected signals must be demodulated using specialized algorithms and specific programs.

The liquid-level measurement method adopted in this study is ultrasonic impedance, which requires the installation of an ultrasonic transducer on one side of the container wall. When the internal medium of the container varies, the attenuation range of ultrasonic echoes on the container wall differs. However, several factors affect the accuracy of ultrasonic sensors: temperature [8], humidity [9], the surface of the object being measured [10], sensor installation position [11], stability of power supply voltage [12], and environmental noise [13], among others. Among them, temperature is considered a crucial influencing factor. Temperature is a critical factor influencing ultrasound measurements. Both the attenuation and speed of sound, as well as the density of the medium, are closely related to temperature [14,15]. Variations in temperature can affect the accuracy of ultrasound measurements [16], thereby reducing the reliability of liquid-level measurements.

Temperature compensation is crucial in various fields, primarily manifesting in several aspects: in many scientific, engineering, and manufacturing applications, temperature variations can affect the accuracy and stability of measurement devices [17]. Through temperature compensation, adverse effects of temperature on measurement results or control systems can be eliminated or reduced, achieving more precise measurement and control. Temperature significantly impacts the performance of many materials [18]. In some applications, such as electronic devices and optical components, temperature fluctuations may lead to variations in material performance. Temperature compensation ensures the stable performance of equipment under different temperature conditions. The performance of many sensors and instruments is greatly affected by temperature variations [19]. Temperature compensation can adjust the output of sensors and instruments through calibration or other methods, ensuring their accuracy and reliability under different temperatures. Overall, temperature compensation helps improve the stability, accuracy, and reliability of measurement, control, and monitoring systems, making it an indispensable key technology in many applications.

From previous research, temperature, as a factor affecting the accuracy of ultrasonic detection, has attracted the attention of many scholars. Lu [20] and colleagues proposed a strategy using diffused ultrasonic waves for temperature compensation. This method achieves a 95% detection rate for damage when the temperature changes by over 30 °C. However, changes in the frequency of ultrasonic signals may affect the compensation effect, and the compensation performance strongly depends on mode purity and signal complexity. Zhan [21] and others proposed a compensation method combining ultrasonic spectroscopy and a temperature compensation model based on Si-PLS, which can compensate for the concentration of multiple-component mixtures in the range of 16 °C to 40 °C. However, bubbles may appear in industrial applications, greatly affecting ultrasonic signals, and the compensation temperature range is not extensive enough. Wang [22] and colleagues proposed a temperature compensation method using adaptive digital filtering and OBS (Optimal Baseline Selection) for ultrasonic-guided waves. This method effectively achieves temperature compensation over a large temperature change range (at least 10 °C), demonstrating outstanding effectiveness and robustness. Wang [23] and others proposed a temperature-load compensation method based on a reference matching tracking algorithm, which improves compensation quality through iterative compensation by designing temperature-load conditions and analyzing the relationship between amplitude, arrival time, and coupling conditions, but it requires a large amount of reference data. Zou [24] discovered that the acoustic impedance of seabed sediments increases with rising temperature, while the impedance ratio and reflection coefficient slightly decrease. Ge [25] proposed a broad-band focusing phenomenon, attributed to the temperature gradient inducing the necessary refractive index in a medium (air) and the continually changing acoustic impedance, which prevents potential impedance mismatches between the lens and air. Marsh [26] suggested that the impedance varies directly with temperature, exhibiting a negative linear relationship, based on the impedance differences between perfluorocarbons, substrates, and propagation media.

Based on the current research status, this study develops a temperature compensation mechanism for an ultrasonic impedance-based point liquid-level detection system. It primarily investigates the impact of temperature-induced drift on the ultrasonic liquid-level detection system, analyzing the relationships between ultrasonic echo energy, temperature, and actual liquid-level height. A novel temperature compensation algorithm is proposed, based on the least squares method, to mitigate or eliminate the effects of temperature. The first section provides an overview of the research status in ultrasonic level measurement technology and temperature compensation techniques. The second section describes the working principle of the ultrasonic sensor and the theoretical understanding of how temperature affects ultrasonic level measurements. The third section introduces the temperature compensation model and explains its contribution to enhancing the accuracy of ultrasonic liquid-level measurements. The fourth section summarizes the main contributions and findings of this study.

## 2. Theoretical Framework and Methodology

The ultrasonic sensor described employs the principle of ultrasonic Lamb waves to detect liquid levels within a container, based on the wave’s transmission and reflection coefficients. A greater difference in acoustic impedance results in reduced transmitted ultrasonic energy and increased reflected energy. The disparity in acoustic impedance between the liquid and air causes a gradual attenuation of the ultrasonic energy as it propagates through the liquid, while the reflected energy is captured by the receiver. Variations in reflection and transmission coefficients affect the attenuation characteristics of the Lamb waves. Specifically, when ultrasonic waves encounter a liquid, reflections and transmissions occur at the container walls and the gas interface. Given that air has a significantly lower acoustic impedance compared to the liquid, the reflection coefficient is higher and the transmission coefficient is lower at the air interface. Consequently, changes in the remaining ultrasonic energy received allow for the determination of the liquid level relative to the probe’s installation position.

### 2.1. Temperature Influence on Ultrasonic Waves

The experiments in this study are based on the velocity and attenuation characteristics of ultrasonic waves. Different temperatures affect the velocity of ultrasonic waves, which in turn influences their attenuation properties. The equation for the influence of temperature on velocity is derived as follows:

For small perturbations of sound waves, their propagation process can be considered an isentropic process, thus we have determined the expression for sound waves [27], as shown in Equation (1):(1)$$c=\sqrt{\frac{dp}{d\rho }}$$

In an isentropic process, the relationship between pressure *p* and density *ρ* is represented by the following Equation (2):(2)$$p\rho^{-\kappa }=\rho^{1-\kappa }RT=C$$

The parameter κ represents the adiabatic index of air, *R* stands for the gas constant of air, *T* denotes temperature, and *C* is a constant. From these parameters, we can derive Equations (3) and (4).
(3)$$\frac{dp}{d\rho }=C\kappa \rho^{\kappa -1}=\kappa RT$$


(4)
$$c=\sqrt{\frac{dp}{d\rho }}=\sqrt{\kappa RT}$$


This indicates that temperature has an impact on the speed of sound in air. For liquid media, Equation (5) proposed in reference [28] shows that the speed of sound in liquids is positively correlated with temperature.
(5)$$w=\sum_{j=0}^{3}b_{j}T$$

Within this context, *w* represents the speed of sound, *T* is the temperature, and *b* is a constant. Meanwhile, attenuation refers to the gradual decrease in the amplitude of a sound wave as it propagates, which is independent of the medium. For liquid media, the primary form of attenuation is absorption [29], and its attenuation coefficient follows the Equation (6):(6)$$\alpha =\alpha_{a}=\frac{8\pi^{2}f^{2}\eta }{3\rho c^{3}}$$

Within this context, *η* represents the viscosity coefficient of the medium. The viscosity coefficient primarily affects the attenuation coefficient, being directly proportional to it, whereas density and sound velocity are inversely proportional to the attenuation coefficient. In this equation, *η*, *ρ*, and *c* all vary with temperature, hence the attenuation coefficient α is influenced by temperature. Additionally, the relationship between temperature and the attenuation coefficient can be derived, denoted by Equation (7).
(7)α=f(T)
When the attenuation is affected, it results in a variation in the ultrasonic energy received by the sensor. According to Equation (8) proposed in reference [30], the relationship between the ultrasonic echo energy and the amplitude is given by:(8)$$E=\frac{1}{2}\rho c^{2}A^{2}f^{2}$$

In this equation, *A* represents the amplitude and *f* is the frequency. According to this formula, when attenuation and sound speed are affected, they lead to different voltage amplitudes, which in turn result in variations in the displayed liquid level, causing experimental errors.

In practical applications, apart from the situations in general media mentioned above, when it comes to the special scenarios where aluminum plates are at the interface with air or liquid, the influence of temperature on the propagation characteristics of acoustic waves presents some unique features. In the scenarios involving the interface between aluminum plates and air or liquid in this paper, the impact of temperature on the group velocity of plate acoustic waves is a key factor, which involves complex changes in multiple aspects such as the material properties of aluminum plates, the acoustic propagation mechanism, and the interaction with adjacent media. The following will elaborate in detail on how temperature affects the group velocity of plate acoustic waves in aluminum plates.

Firstly, we will discuss how temperature affects the material properties of aluminum plates. When the temperature rises, the Young’s modulus *E* [31] of the aluminum plate will decrease. Generally, the relationship of its change can be approximately expressed as:(9)$$E=E_{0}(1-\alpha T)$$
where *E*_0_ is the initial Young’s modulus and *α* is the temperature coefficient. Young’s modulus is an important parameter that describes the ability of a material to resist elastic deformation, and the change in its value directly affects the propagation characteristics of acoustic waves in the aluminum plate. As the temperature increases, the binding force between atoms weakens, resulting in the material being more prone to elastic deformation and thus a decrease in Young’s modulus. Such a change will have an impact on the propagation speed of acoustic waves and, in turn, on the group velocity.

Besides Young’s modulus, Poisson’s ratio [32] will also change with temperature and can usually be expressed as:(10)$$\nu =\nu_{0}+\beta T$$
where *v*_0_ is the initial Poisson’s ratio and *β* is the temperature coefficient of Poisson’s ratio. When Poisson’s ratio changes, the deformation mode of the aluminum plate during the propagation of acoustic waves will also change accordingly, affecting the propagation path and speed of acoustic waves and ultimately acting on the group velocity.

Secondly, we will discuss the influence of temperature on the propagation speed of acoustic waves in aluminum plates. According to the theory of elastic waves, the propagation speed of acoustic waves in solid media is related to the elastic constants of the material (such as Young’s modulus and Poisson’s ratio). In aluminum plates, the longitudinal wave velocity *C_L_* and the transverse wave velocity *C_T_* are expressed as:(11)$$c_{L}=\sqrt{\frac{\lambda +2\mu }{\rho }}$$
(12)$$c_{T}=\sqrt{\frac{\mu }{\rho }}$$
where *λ* and *μ* are the Lamé constants, which are related to Young’s modulus and Poisson’s ratio. Among them, *λ* and *μ* are expressed as:(13)$$\lambda =\frac{\nu E}{\left(1+\nu )(1-2\nu \right)}$$
(14)$$\mu =\frac{E}{2(1+\nu )}$$

Due to the changes in Young’s modulus and Poisson’s ratio caused by temperature, both the longitudinal wave velocity and the transverse wave velocity will change.

Finally, we will discuss the influence of temperature on the acoustic coupling at the interface between aluminum plates and air or liquid. The acoustic impedance difference between aluminum plates and air or liquid is relatively large, and temperature changes will further affect this difference. The acoustic impedance *Z* is expressed as:(15)Z=ρc
where *c* is the sound velocity and *ρ* is the material density. Since temperature affects the sound velocity in aluminum plates as well as the density of the aluminum plates themselves, their acoustic impedance will change. When the temperature changes, the acoustic impedance matching situation at the interface between the aluminum plate and the adjacent medium (air or liquid) will also change. At the interface, the energy of acoustic waves will be distributed according to the acoustic impedance matching situation. Part of the energy will be reflected back to the aluminum plate, and part of the energy will be transmitted into the adjacent medium. The change in acoustic impedance caused by temperature changes will lead to a change in the energy distribution at the interface. When the reflected energy increases, the energy of acoustic waves propagating in the aluminum plate decreases, which will affect the propagation speed and group velocity of acoustic waves.

In conclusion, temperature has a complex and significant impact on the group velocity of plate acoustic waves at the interface between aluminum plates and air or liquid through multiple ways, such as changing the material properties of aluminum plates, affecting the propagation speed of acoustic waves in aluminum plates, and altering the acoustic coupling situation at the interface between aluminum plates and air or liquid.

### 2.2. Temperature Compensation Algorithm

Based on the above discussion, it can be observed that the output voltage at the sensor end varies with temperature. This paper proposes a compensation algorithm based on the least squares method [33]. When the temperature effect is removed, the output is a function of voltage and liquid level, as shown in Equation (16) below. However, when temperature affects the system, the specific liquid level is related to the voltage and attenuation coefficient, as expressed in Equation (17) below.
(16)$$h_{0}=f(u_{f})$$


(17)
$$h_{1}=g(u_{h},\alpha )$$


The peak-to-peak value of the output signal when *u_f_* serves as the reference signal, *u_h_* represents the peak-to-peak value of the output signal under temperature influence, and α represents the attenuation coefficient. Equation (18) can be derived from Equations (16) and (17).
(18)$$h=g(u_{h},\alpha )=g[u_{h},f(T)]=F(u_{h},T)$$

The equation above represents the relationship between liquid-level height, output voltage peak-to-peak value, and temperature. However, to improve measurement accuracy, temperature compensation is necessary. We employ a three-dimensional regression method to establish the correspondence between liquid-level height, output voltage peak-to-peak value, and temperature. The coefficients of the regression equation are fitted using the least squares method.

The regression equation is as follows:(19)$$\begin{array}{l} h=c_{00}+c_{01}u_{h}+c_{10}T+c_{02}u_{h}^{2}+c_{11}u_{h}T+c_{20}T^{2}\\ +c_{03}u_{h}^{3}+c_{12}u_{h}^{2}T+c_{21}u_{h}T^{2}+c_{30}T^{3} \end{array}$$

In order to obtain the coefficients *c_i,j_* (*i* + *j* < 3)
(20)$$Q={\left(h-h_{0}\right)}^{2}=min$$

So,
(21)$$Q=\sum_{k=0}^{m\times n}\begin{array}{l} (c_{00}+c_{01}u_{h}+c_{10}T+c_{02}u_{h}^{2}+c_{11}u_{h}T+c_{20}T^{2}\\ +c_{03}u_{h}^{3}+c_{12}u_{h}^{2}T+c_{21}u_{h}T^{2}+C_{30}T^{3}-h_{0k}{)}^{2} \end{array}$$

The variable *h*_0*k*_ denotes the actual liquid-level height, *k* = 0,1,2…*m* × *n*

By setting the partial derivatives of each term to 0, the values of the coefficients are determined.

Thus, we have:
(22)$$\frac{𝜕Q}{𝜕c_{i,j}}=0,(i+j<3)$$

According to the Equations (21) and (22), we can derive Equation (A1). Equation (A1) is shown in the Appendix A.

According to Equation (22), the coefficients *c_i,j_* can be quickly solved, thereby obtaining the regression equation. Consequently, the specific liquid-level height is obtained. In practical measurements, if the corresponding voltage amplitude under a given temperature is obtained, the actual compensated liquid-level height can be obtained according to Equation (19).

### 2.3. Experimental Platform Design

Building upon the theoretical foundation outlined above, to accurately measure liquid levels without being influenced by acoustic interference and attenuation within the liquid, we established an external fixed-point non-destructive monitoring system for liquid levels. The experimental platform design diagram and measurement platform are shown in Figure 1, and the parameters of the measurement equipment are presented in Table 1. This system mainly consists of two sensors and an external container. Both sensors are piezoelectric ceramic (PZT) chips, one serving as the transmitter and the other as the receiver. Considering the widespread use of aluminum alloys in aerospace applications, we selected aluminum alloy as the material for the test container in our experiments. The container has a wall thickness of 3 mm, and its internal medium is water and air. Due to the excellent stability of silicone gel, which maintains a high acoustic transmission efficiency, silicone is used as a coupling agent to connect the transducer to the water tank. When ultrasonic waves reach the interface between the container’s inner wall and the internal medium, refraction and transmission phenomena occur. By installing the sensor vertically on the outside of the container and positioning the receiver at the same column position, we can accurately measure changes in ultrasonic echo energy to determine the liquid-level height between the two sensors.

Firstly, we utilize the high-speed operational amplifier AD603 (Analog Devices, Inc., Norwood, MA, USA) along with its associated circuitry to generate square wave signals with a defined pulse width. This signal drives the ultrasonic transducer to emit a continuous burst of ultrasonic waves every 25 ms. The received ultrasonic signals at the receiving end are then used to facilitate signal transmission. Subsequently, analog data are displayed using an oscilloscope (TDS 1001B, Tektronix, Shanghai, China), while digital quantities are captured by a data collector (USB-1610, New Super Technology, Beijing, China). The final data are then displayed by relevant software on a computer. In the experiments, the entire detection system was placed inside a high- and low-temperature chamber, where it was exposed to various temperature environments. Temperature readings were recorded using a digital probe thermometer (TP101, Shengce, Wenzhou, China). Water was gradually added to the container, with the lower-mounted sensor as the origin, and the output signal amplitude collected by the data acquisition card was recorded at liquid-level heights ranging from 0 cm (liquid-level below the lower sensor) to 10 cm (liquid-level above the upper sensor). Measurements were taken every 1 cm, with three sets of data recorded each time and subjected to sliding average filtering. The lower sensor was set as the transmitting end and the upper sensor as the receiving end, and multiple measurements were repeated. Throughout the experiment, the oscilloscope probe was primarily connected to the receiver end (Rx) of the ultrasonic level sensor. The voltage amplitude displayed on the oscilloscope represents the received ultrasonic energy. To minimize errors from this part, the average value was obtained by taking multiple measurements. Additionally, efforts were made to eliminate other external interferences during the experiment, such as electromagnetic interference between devices and environmental noise.

## 3. Results and Discussion

To assess the impact of temperature on the entire experiment, this section conducted grouped experiments based on the aforementioned theory and experimental setup, discussing the relationship between temperature, ultrasonic echo energy, and liquid-level height.

### 3.1. Relationship Between Temperature, Ultrasonic Echo Energy, and Liquid-Level Height

Two sensors were mounted on the outer side of an aluminum alloy container with a wall thickness of 3 mm, enabling external apex-style non-intrusive liquid-level measurement. Initially, at a room temperature of 20 °C, ultrasonic echo energy for various liquid-level heights was measured. Three sets of experiments were conducted, and the echo amplitudes of the sensors were recorded, with the average values calculated. Subsequently, the relationship between ultrasonic echo energy and liquid-level height at room temperature was fitted, yielding a relative error of 5.72%. As the liquid level increased, the ultrasonic echo energy gradually decreased, consistent with theoretical expectations. Figure 2 illustrates this phenomenon.

Subsequently, we conducted multiple experiments to measure liquid levels at different temperatures. It can be clearly observed from Figure 3 that under different temperatures, the ultrasonic echo energy and the liquid-level height approximately exhibit a linear relationship. As the liquid-level height gradually increases, the ultrasonic echo energy shows a downward trend. This trend is consistent with theoretical expectations. As the liquid level rises, the propagation path of ultrasonic waves in the liquid becomes longer and the energy attenuation increases, resulting in a reduction in the received echo energy. We can see from the figure the trend of the ultrasonic echo energy changing with the liquid-level height. This indicates that the influence of the liquid-level height on the ultrasonic echo energy is relatively significant, and a small change in the liquid level will cause a relatively large change in the echo energy. By comparing the curves under different temperatures, it can be seen that the influence of temperature on the relationship between the ultrasonic echo energy and the liquid-level height is nonlinear. Figure 4 shows that for different liquid-level heights (from 0 cm to 10 cm), the change of the ultrasonic echo energy with temperature presents a certain regularity. Generally speaking, at the same liquid-level height, the ultrasonic echo energy increases as the temperature rises. Taking the liquid-level height of 0 cm as an example, as the temperature gradually increases from 0 °C to 40 °C, the ultrasonic echo energy shows a gradually rising trend, and the rising amplitude is relatively obvious. This is because when the temperature rises, the propagation speed of ultrasonic waves in the medium accelerates and the attenuation coefficient decreases, thus leading to an increase in the received echo energy. However, when the temperatures were 0 °C, 10 °C, 30 °C, and 40 °C, if the fitting equation at room temperature was continued for data processing, the relative errors of the fitted data were 38.42%, 16.06%, 27.45%, and 37.16%, respectively. Thus, the average relative error of the fitted data at room temperature was 24.962%. Consequently, the relationship between the output ultrasonic echo energy and liquid-level height cannot be represented by a fixed equation. We can conclude that temperature variations significantly affect the measurement of liquid-level height using ultrasound, thereby greatly reducing measurement accuracy.

### 3.2. Temperature Compensation Model

Figure 5 presents the results of compensation using the least squares fitting surface. Table 2 shows the parameter values of Equation (12), and Table 3 displays the goodness-of-fit parameters for the polynomial, with a determination coefficient of 0.99144. By substituting the actual measured ultrasonic echo energy and the current temperature into the equation, the compensated liquid-level height was obtained. A comparison between the actual and compensated liquid-level heights is depicted in Figure 6. Figure 7 shows the actual relative error after temperature compensation, with an average relative error of 3.427% and a maximum error of 0.336 cm. The maximum absolute error is only 11.2%, corresponding to a maximum error of 0.336 cm. Compared to the 24.962% relative error of the fitting results before compensation mentioned in Section 3.1, the temperature compensation model has reduced the average relative error by 21.535%.

To improve the reliability of the fitting model, random temperature verification was carried out for the liquid-level height ranging from 0 to 10 cm. Compared with the temperatures used for polynomial coefficient fitting, four random temperatures, namely 5 °C, 17 °C, 38 °C, and 47 °C, were selected, and the ultrasonic echo amplitudes at these temperatures were measured respectively. These ultrasonic echo amplitudes were substituted into Equation (19) for verification, and the fitted liquid-level heights at each point were obtained. Through error analysis of these values, the relative errors under the four temperature conditions were calculated to be 3.763%, 5.143%, 4.072%, and 3.08%, respectively. The experimental results are shown in Table 4.

At room temperature, water was randomly added to or drained from the container, and the actual liquid-level values were read using a ruler. Seven randomly selected liquid-level values were tested, and the recorded ultrasonic echo energy and ambient temperature were inputted into Equation (19) to calculate the liquid-level values. A comparison was made between the actual liquid-level values and the calculated ones to analyze the range of errors. Table 5 illustrates this information.

It can be known from the above results that the maximum detection error is 0.19 cm, and the average relative error is 2.12%. Reference [34] employed a standard test sample on the pipe surface to calculate temperature changes using ultrasonic signal propagation delay (DT), establishing an error model for temperature and flow relationships. Their temperature compensation method achieved a maximum relative error of 8.92%, which is higher than the 3.427% average relative error of the proposed temperature compensation method in this study, reducing the error by 5.493%. This validates that the proposed temperature compensation method effectively minimizes environmental temperature drift, enhancing measurement accuracy.

In the subsequent experiments, to ensure the rigor of the experiments, we adjusted the environmental temperature and set it to 0 °C, 5 °C, 10 °C, 17 °C, 30 °C, 38 °C, 40 °C, and 47 °C respectively for conducting the experiments. Then, we measured the ultrasonic echo amplitudes for these seven randomly selected liquid-level values. After that, these measured ultrasonic echo amplitudes and the corresponding temperature values were substituted into Equation (19) for verification, and error analysis was performed on them. The experimental results are shown in Table 6 and Table 7.

It can be clearly seen from the above experimental results that under the conditions of random temperatures and random liquid-level heights, the relative error values of the liquid-level heights obtained by fitting through this formula are all less than 5%. This result fully verifies that the temperature compensation method proposed in this paper can indeed effectively reduce the impact brought by the environmental temperature drift and, consequently, significantly improve the accuracy of liquid-level measurement.

## 4. Conclusions

Based on the principles of ultrasonic non-destructive testing, we established an external fixed-point liquid-level detection system experimental platform. This study primarily discusses the influence of temperature changes on liquid-level measurement results and proposes a compensation method to mitigate the effects of temperature drift. Theoretical analysis indicates that an increase in temperature causes changes in the speed and attenuation of sound waves, thereby affecting the signal output. Environmental temperature fluctuations significantly impact the accuracy of the ultrasonic impedance method for measuring liquid levels. Measurement results show that ultrasonic echo energy increases with rising environmental temperature. Therefore, a temperature compensation method based on the least squares method is proposed. This method fits temperature, ultrasonic echo energy, and liquid-level height through least squares fitting to reduce the impact of temperature drift. Through comparative analysis, the average relative error of the temperature compensation method proposed in this paper is 3.427%, the error range is 0.03 cm to 0.336 cm, and the average relative error is reduced by 21.535%, which verifies the effectiveness of the method. Based on the above experimental content, through in-depth experiments and analyses of the liquid-level measurement system under different temperature conditions, the temperature compensation range of the model has been determined. The relative error is taken as the key indicator to define the temperature compensation range. When the relative error is within an acceptable range, it indicates that the model can effectively compensate for the impact of temperature drift on liquid-level measurement. By comprehensively considering the data at various temperature points, it can be known that within the temperature range from 0 °C to 50 °C, the relative errors of the model for measuring different liquid-level heights can basically be controlled within a relatively small range (most relative errors are less than 5%). This means that within this temperature range, the model has good temperature compensation ability and can measure the liquid-level height relatively accurately, meeting the requirements for liquid-level measurement accuracy in industrial applications. Our work demonstrates that this method effectively compensates for temperature variations, thus holding significant practical value for improving the accuracy of ultrasonic-based measurements in industrial settings.

## Figures and Tables

**Figure 1 micromachines-16-00024-f001:**
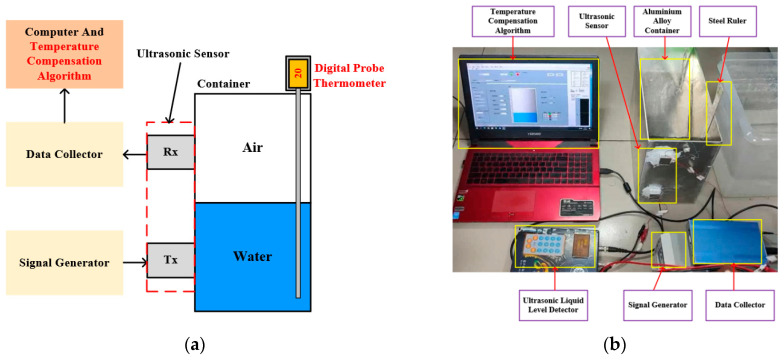
Liquid-level measuring system: (**a**) Experimental platform design diagram.; (**b**) Experimental platform design diagram.

**Figure 2 micromachines-16-00024-f002:**
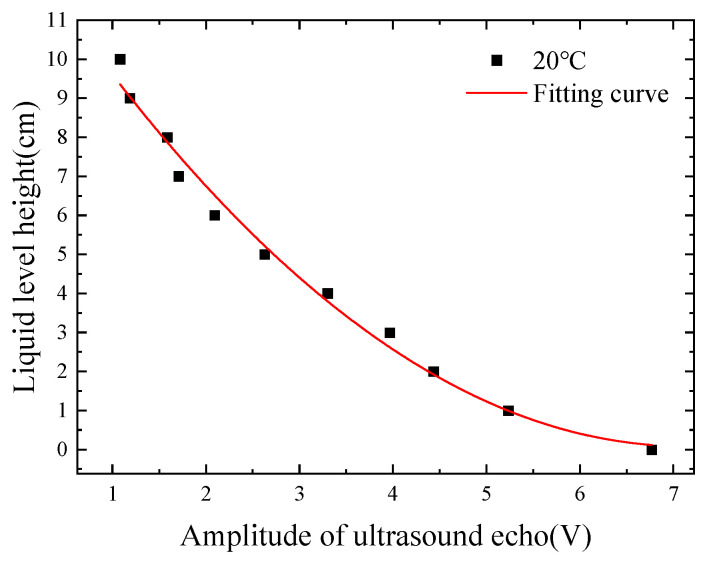
The relationship between ultrasonic echo energy and liquid-level height at 20 °C.

**Figure 3 micromachines-16-00024-f003:**
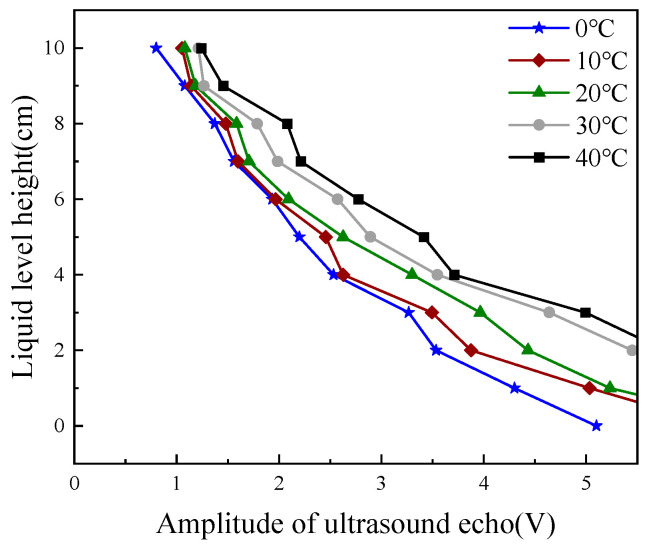
The relationship between ultrasonic echo energy and liquid-level height at different temperatures.

**Figure 4 micromachines-16-00024-f004:**
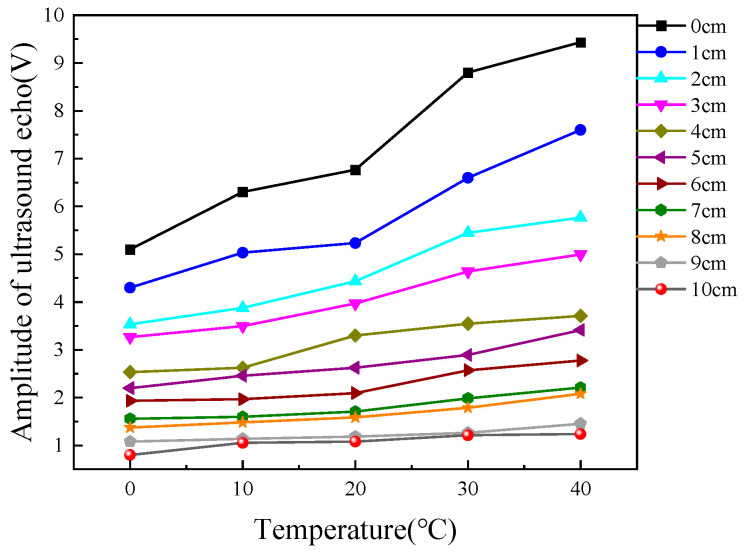
The relationship between ultrasonic echo energy and temperature.

**Figure 5 micromachines-16-00024-f005:**
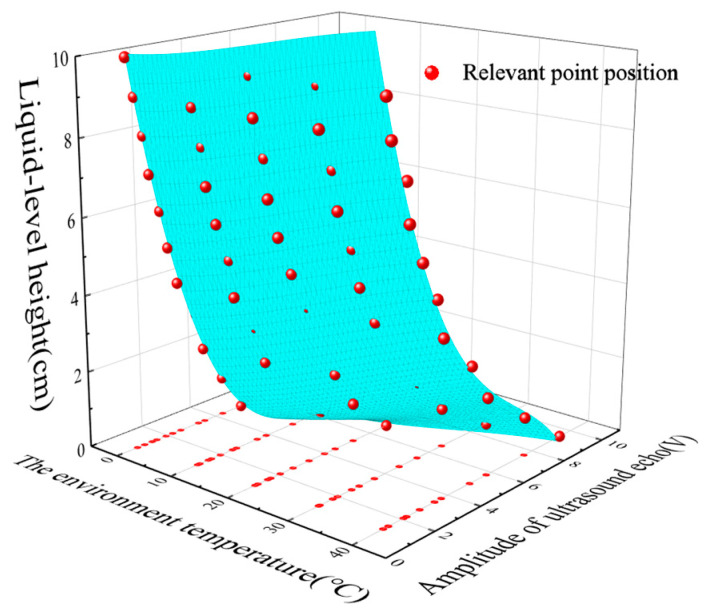
Temperature compensation model.

**Figure 6 micromachines-16-00024-f006:**
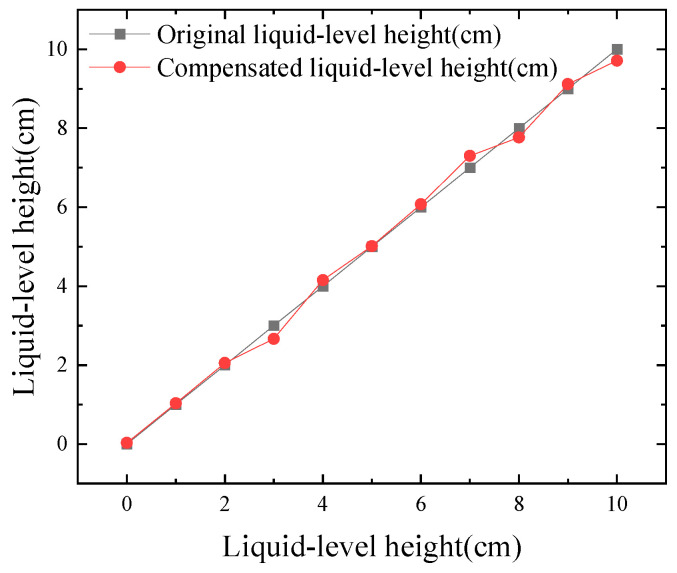
Compare the compensated liquid-level height with the original liquid-level height.

**Figure 7 micromachines-16-00024-f007:**
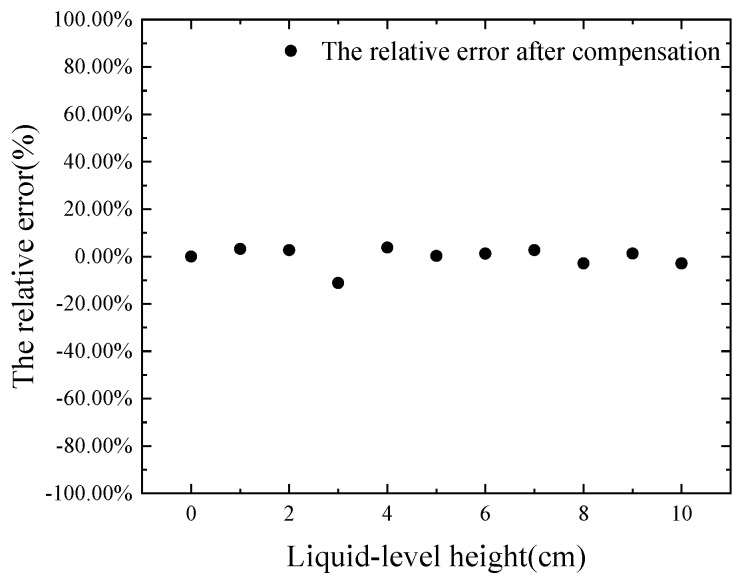
The relative error after compensation.

**Table 1 micromachines-16-00024-t001:** Experimental parameters and initial values.

Symbol	Specification	Initial Values
c	Velocity of ultrasound	2775 m/s
Am	Ultrasonic amplitude	±15 V
T	Experimental temperature	0 °C, 10 °C, 20 °C, 30 °C, 40 °C

**Table 2 micromachines-16-00024-t002:** Function parameter value.

Parameter in the Formula	Numerical Value	95% Confidence Interval
*C* _00_	13.84612	(13.153814, 14.538426)
*C* _01_	−5.18764	(−5.447022, −4.928258)
*C* _10_	2.84387 × 10^−4^	(2.7016765 × 10^−4^, 2.9860635 × 10^−4^)
*C* _02_	0.61793	(0.5870335, 0.6488265)
*C* _11_	0.01859	(0.0176605, 0.0195195)
*C* _20_	0.00114	(0.001083, 0.001197)
*C* _03_	−0.02533	(−0.0265965, −0.0240635)
*C* _12_	−0.00184	(−0.001932, −0.001748)
*C* _21_	−4.42907 × 10^−6^	(−4.6505235 × 10^−6^, −4.2076165 × 10^−6^)
*C* _30_	−1.6108 × 10^−5^	(−1.69134 × 10^−5^, −1.53026 × 10^−5^)

**Table 3 micromachines-16-00024-t003:** Polynomial fit.

Parameter in the Formula	Numerical Value
Reduced Chi-square	0.08717
R-square	0.99287
Adjusted R-square	0.99144

**Table 4 micromachines-16-00024-t004:** Fitted liquid-level height under random temperature conditions.

H (cm)	5 °C (V)	H_1_ (cm)	17 °C (V)	H_2_ (cm)	38 °C (V)	H_3_ (cm)	47 °C (V)	H_4_ (cm)
0	5.7	−0.08	6.58	0.22	9.1	0.074	9.53	−0.16
1	4.5	0.97	5.15	1.1	7.53	1.08	8.13	1.07
2	3.67	2.12	4.13	2.19	5.67	2.17	6.53	2.08
3	3.33	2.72	3.77	2.73	4.79	2.91	5.2	3.03
4	2.59	4.32	3.03	4.02	3.67	4.28	4.2	4.04
5	2.33	4.98	2.55	5.07	3.27	4.92	3.67	4.75
6	1.94	6.1	2.07	6.31	2.7	5.99	2.83	6.15
7	1.58	7.24	1.68	7.45	2.17	7.21	2.34	7.17
8	1.43	7.75	1.58	7.77	1.89	7.92	2.11	7.71
9	1.1	8.97	1.18	9.14	1.43	9.24	1.51	9.31
10	0.9	9.76	1.08	9.5	1.22	9.9	1.25	10.08
Relative error (*δ*)		3.763%		5.143%		4.072%		3.08%

H: Actual liquid level; H_1_: The liquid-level height was fitted at 5 °C; H_2_: The liquid-level height was fitted at 17 °C; H_3_: The liquid-level height was fitted at 38 °C; H_4_: The liquid- level height was fitted at 47 °C.

**Table 5 micromachines-16-00024-t005:** Error analysis of seven sets of data.

H (cm)	V (V)	H_1_ (cm)	δ_1_ (%)	H_2_ (cm)	δ_2_ (%)	H_3_ (cm)	δ_3_ (%)
4.8	3.168	4.06	15.4	5.35	10.28	4.61	3.96
5.7	2.712	5.03	11.8	6.02	5.32	5.51	3.33
6.3	2.436	5.67	10	6.50	3.06	6.13	2.70
7.4	1.974	6.82	7.8	7.50	1.30	7.30	1.35
8.1	1.704	7.55	6.8	8.16	0.76	8.06	0.49
8.9	1.458	8.24	7.4	8.97	0.81	8.81	1.01
9.4	1.332	8.60	8.5	9.48	0.98	9.21	2.02

H: Actual liquid level; V: Amplitude of ultrasound echo; H_1_: Curve fitting liquid level; H_2_: Fitted liquid levels in reference [34]; H_3_: Polynomial fitting of liquid level; δ_1_, δ_2_, δ_3_: Relative error.

**Table 6 micromachines-16-00024-t006:** Fitted level heights of seven groups of random liquid levels (0–47 °C).

H (cm)	0 °C	H_1_	5 °C	H_2_	10 °C	H_3_	17 °C	H_4_	30 °C	H_5_	38 °C	H_6_	40 °C	H_7_	47 °C	H_8_
4.8	2.31	4.85	2.4	4.81	2.5	4.79	2.63	4.88	3.14	4.66	3.31	4.85	3.53	4.61	3.67	4.74
5.7	2	5.74	2.13	5.54	2.2	5.57	2.32	5.64	2.65	5.65	2.83	5.73	2.97	5.57	3.06	5.73
6.3	1.81	6.33	1.9	6.22	1.91	6.4	2.08	6.28	2.42	6.17	2.58	6.25	2.65	6.21	2.76	6.29
7.4	1.45	7.55	1.54	7.38	1.61	7.34	1.7	7.39	1.89	7.53	2.03	7.56	2.12	7.42	2.25	7.38
8.1	1.26	8.24	1.34	8.08	1.39	8.08	1.52	7.96	1.71	8.04	1.84	8.06	1.84	8.15	1.96	8.08
8.9	1.1	8.85	1.12	8.89	1.22	8.69	1.26	8.84	1.41	8.96	1.52	8.98	1.55	8.96	1.64	8.93
9.4	0.9	9.66	0.95	9.56	1.05	9.33	1.11	9.38	1.23	9.55	1.37	9.43	1.33	9.63	1.47	9.42

H: Actual liquid level; H_1_ (cm): The liquid-level height was fitted at 0 °C; H_2_ (cm): The liquid-level height was fitted at 5 °C; H_3_ (cm): The liquid-level height was fitted at 10 °C; H_4_ (cm): The liquid-level height was fitted at 17 °C; H_5_ (cm): The liquid-level height was fitted at 30 °C; H_6_ (cm): The liquid-level height was fitted at 38 °C; H_7_ (cm): The liquid- level height was fitted at 40 °C; H_8_ (cm): The liquid-level height was fitted at 47 °C.

**Table 7 micromachines-16-00024-t007:** Error analysis of 7 sets of data (0–47 °C).

Temperature	0 °C	5 °C	10 °C	17 °C	30 °C	38 °C	40 °C	47 °C
Relative error (*δ*)	2.02%	2.39%	2.64%	2.57%	2.4%	2.11%	3.48%	4.06%

## Data Availability

The original contributions presented in the study are included in the article, further inquiries can be directed to the corresponding author.

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
