# Peer review of "A Novel Temperature Drift Compensation Algorithm for Liquid-Level Measurement Systems"

_micromachines, 2024, doi:10.3390/mi16010024_

Round 1

Reviewer 1 Report (Previous Reviewer 1)

Comments and Suggestions for Authors

Unfortunately, the changes that were made to the text of the work does not eliminate the drawbacks noted in my previous review. The entire theoretical framework from Section 2.1 still applies to ultrasound in fluid medium (liquid or gas). However, the experimental part of the work uses acoustic waves in a 3 mm thick aluminum plate (solid medium). From the description in Section 2.3, I can conclude that bursts of ultrasonic waves are emitted and received (at what frequency?), so we should operate with the group velocity of plate acoustic waves in aluminum bordered with air or liquid. In general, the presented work has a good part (description of the experiment: Section 2.3 and the experimental results obtained: Figures 2, 3, 4). However, the quality of Sections 2.1, 2.2 and 3.2 does not allow me to accept this work. I can advise the authors to ask for help from more experienced colleagues for a complete rewrite of the theoretical part of the work and better implementation (and justification!) of the thermal compensation algorithm.

Author Response

Reviewer 2 Report (Previous Reviewer 2)

Comments and Suggestions for Authors

1. What are the reasons why you used only three data sets for each measurement point? How does this affect the robustness of the conclusions?

2. Why did you choose the least squares method for the temperature compensation model? Were other modeling methods considered?

3. How can Figure 7 be interpreted in the context of industrial applications? Can this model be extended to other temperature ranges or different environments?

4. Can this temperature compensation model be applied to other liquid media (eg liquids with variable viscosity)? If not, what adjustments would be needed?

5. What are the main limitations of this algorithm in extreme conditions such as temperatures below 0°C or above 40°C?

Author Response

Reviewer 3 Report (New Reviewer)

Comments and Suggestions for Authors

This paper discusses the effects of temperature changes on liquid-level measurement results and a novel compensation method to mitigate temperature drift. The proposed method is based on the least squares method for temperature compensation, which uses least squares fitting to model temperature, ultrasonic echo energy, and liquid-level height to reduce the impact of temperature drift. Leveraging an external fixed-point liquid-level detection system experimental platform, the impact of temperature drift on ultrasonic echo energy and actual liquid-level height is examined. This method effectively compensates for temperature variations, which helps realize the significant practical value for improving the accuracy of ultrasonic-based measurements in industrial settings. The article demonstrates strong innovation and practical applicability, with clear and fluent writing and significant practical value for improving the accuracy of industrial ultrasonic measurements. However, there are a few issues that need to be clarified. I suggest making some minor revisions with the following comments.

1. Pay attention to the clarity of the images used in this paper (such as Figure 1 and Figure 5). The font size of the axis labels and legends in the figures is too small and difficult to read. The font size of the article in the figures should be roughly consistent with the font size of the figure numbers in the main text.

2. The font size of the text in the tables should be consistent. The font size in Table 4 seems to be slightly larger than that in Tables 2 and 3. Additionally, the size of the images in the article can be appropriately increased to make the images and tables look more coordinated.

3. To avoid ambiguity, the subscripts of C00-C30 in Table 2 should be consistent with the subscripts of parameters such as C00 in equations (12) and (14).

4. A more detailed description of the content and trends shown in Figures 3 and 4 should be provided before the conclusion in line 260.

Author Response

Reviewer 4 Report (New Reviewer)

Comments and Suggestions for Authors

The paper introduces a novel temperature drift compensation algorithm to address the impact of temperature variations on ultrasonic liquid-level measurement systems. It investigates how temperature-induced changes in ultrasonic wave velocity and attenuation affect measurement accuracy. A temperature compensation method is developed using the least squares regression approach, which aims to minimize the errors caused by temperature fluctuations. Experimental validation shows that this method reduces the relative error from 24.96% (before compensation) to 3.43%, demonstrating its effectiveness in enhancing the precision of ultrasonic-based liquid-level measurements. However, before publication, it is necessary to explain or add some content to make the paper complete.

1.      In equation (12), the term of "C12un2T" is incorrect and should be "C12uh2T".

2.      Is the "," symbol after each equation necessary? If it is not necessary, it is recommended to remove the "," symbol after all equations. This is because in some equations, this symbol can cause misunderstandings. For example, in equation (14), this symbol is easily used as a derivative symbol.

3.      The temperature compensation method proposed by the authors is based on data at 0 ℃, 10 ℃, 20 ℃, 30 ℃, and 40 ℃, and uses polynomial fitting to obtain the coefficients of the polynomial. Due to the uniform distribution of 5 temperature points in the fitted data, the accuracy of the model needs to be rigorously tested. The model validation results presented by the authors in Table 4 show that when the temperature is 20 ℃, the prediction error of the model is less than 5% at different liquid level heights. However, the verification process is too simple. In addition to verifying the accuracy of the model at 20 ℃, it is recommended to verify the measurement results and errors of the model at other temperatures.

4.      When the authors verified the accuracy of the model, the selected temperature was the same as the temperature used for polynomial fitting, which was 20 ℃. When verifying the polynomial fitting results, it is recommended to supplement the model measurement results and errors at random temperatures, such as 5 ℃, 17 ℃, 38 ℃, 47℃, etc. These temperatures should be different from the temperature of the input data used to fit the polynomial coefficients

5.      The authors fitted the coefficients of a polynomial using test results from 0 to 50 ℃. When using this system in practical engineering, the errors may vary greatly at different temperatures. It is recommended to supplement the error distribution of the model measurement results at different temperatures and analyze them.

6.      It is recommended to provide the temperature compensation range of the model based on the error distribution at different temperatures as suggested in suggestion (5).

Round 2

Reviewer 4 Report (New Reviewer)

Comments and Suggestions for Authors

The author describes the doubts I have given, and I have no more comments.  

This manuscript is a resubmission of an earlier submission. The following is a list of the peer review reports and author responses from that submission.

Round 1

Reviewer 1 Report

Comments and Suggestions for Authors

This work is devoted to the problem of determining the liquid level in closed vessels using ultrasonic waves propagating along the vessel wall. The idea of this method is not new [V.E. Sakharov et al. Liquid level sensor using ultrasonic Lamb waves, and many others]. The novelty of this work could lie in the proposal of a new method for accounting of temperature drift. However, the implementation of this idea has numerous drawbacks.
1. The theoretical framework of this work is completely irrelevant. The discussion in Section 2.1 relates to the dependence of the speed of sound in air on temperature. However, this work uses acoustic waves in a thin solid shell (vessel wall), which borders on a medium with different acoustic impedance (air and liquid). The theoretical description of the dependence of the speed of sound on temperature is different in this case. Further in the work, the output signal value is used, and not the phase velocity of the wave.
2. The text of the paper widely uses the concept of Ultrasonic Echo Energy (for example, in the title 3.1 on page 6, line 230), but the amplitude of the output electrical signal in volts is used in the calculations. It is known that the energy of an acoustic wave is proportional to the square of the oscillatory velocity of this wave.
3. In the description of the Temperature Compensation Algorithm, only 10 coefficients cij are used. This means that the condition on page 4, line 173 and in Equation (13) for indices i, j should look like i + j <= 3.
4. The formula (10) used for the liquid level height h, contains dependencies on u and T in various combinations. However, as follows from Table 2 that the coefficients C10, C20, C21 and C30 are determined with insufficiently high accuracy so that even their sign can be different. This means that the influence of T, T^2, U*T^2 and T^3 on the result is insignificant. The main contribution is made by the quantities U, U^2, U^3, T*U and T*U^2, i.e. temperature affects the result linearly. Therefore, formula (10) looks overcomplicated to describe this phenomenon.
5. The authors do not write anything about the method of measuring the output signal magnitude and its error. As follows from the description of Figure 1, the signal from the Rx receiver is used, i.e. the magnitude of the transmitted signal is measured, however, in Figures 2, 3 and 4 the authors talk about the magnitude of the sound echo amplitude, i.e. the magnitude of the reflected signal.
All this means that the authors of the work do not quite correctly understand the physics of the phenomenon they use. Therefore, I am forced to recommend rejecting this work. The authors should either more correctly and accurately describe the essence of the phenomenon (make the work more fundamental), or focus on significantly improving the characteristics (for example, accuracy) of the method they propose.

Reviewer 2 Report

Comments and Suggestions for Authors

1. What are the major limitations of the proposed method and how do they influence the results?

2. What difficulties did you encounter in the experimental implementation and how did you manage them?

3. In what practical applications would this thermal drift compensation algorithm be most useful? Have you already tested the system in a real industrial environment?

4. How was the least squares compensation method selected compared to other available methods? Have you evaluated other methods?

5. What are the main limitations of the proposed compensation algorithm? Are there scenarios where the method would not be applicable or effective?

6. To what extent is the performance of the method influenced by extreme temperature conditions or the presence of other disturbing factors such as ambient noise?

7. In figure 1 the design of the experimental platform is shown, but it is not clear how possible external interferences were minimized. Can you explain in more detail how the isolation from background noise and other acoustic disturbances was ensured?

8. You mentioned that temperature significantly influences ultrasound measurements, but you did not elaborate on possible scenarios where the proposed model might not be effective. Can you discuss more about the limitations of the algorithm and the conditions under which it might not work correctly?

9. Details of the materials used, such as silicone coupling gel, are only briefly mentioned. Can you elaborate more on why this material was selected and if there might be more effective alternatives?
